# The Potential of Computational Modeling to Predict Disease Course and Treatment Response in Patients with Relapsing Multiple Sclerosis

**DOI:** 10.3390/cells9030586

**Published:** 2020-03-01

**Authors:** Francesco Pappalardo, Giulia Russo, Marzio Pennisi, Giuseppe Alessandro Parasiliti Palumbo, Giuseppe Sgroi, Santo Motta, Davide Maimone

**Affiliations:** 1Department of Drug Sciences, University of Catania, 95125 Catania, Italy; giulia.russo@unict.it; 2Department of Mathematics and Computer Science, University of Catania, 95125 Catania, Italy; mpennisi@dmi.unict.it (M.P.); giuseppe.parasilitipalumbo@phd.unict.it (G.A.P.P.); giuseppe.sgroi@unict.it (G.S.); 3National Research Council of Italy, 00185 Rome, Italy; s.motta@iac.cnr.it; 4Multiple Sclerosis Center, Neurology Unit, Garibaldi Hospital, 95124 Catania, Italy; dmaimone@tiscali.it

**Keywords:** computational modeling, agent-based modeling, systems biology, multiple sclerosis, immunity, degenerative disease

## Abstract

As of today, 20 disease-modifying drugs (DMDs) have been approved for the treatment of relapsing multiple sclerosis (MS) and, based on their efficacy, they can be grouped into moderate-efficacy DMDs and high-efficacy DMDs. The choice of the drug mostly relies on the judgment and experience of neurologists and the evaluation of the therapeutic response can only be obtained by monitoring the clinical and magnetic resonance imaging (MRI) status during follow up. In an era where therapies are focused on personalization, this study aims to develop a modeling infrastructure to predict the evolution of relapsing MS and the response to treatments. We built a computational modeling infrastructure named Universal Immune System Simulator (UISS), which can simulate the main features and dynamics of the immune system activities. We extended UISS to simulate all the underlying MS pathogenesis and its interaction with the host immune system. This simulator is a multi-scale, multi-organ, agent-based simulator with an attached module capable of simulating the dynamics of specific biological pathways at the molecular level. We simulated six MS patients with different relapsing–remitting courses. These patients were characterized based on their age, sex, presence of oligoclonal bands, therapy, and MRI lesion load at the onset. The simulator framework is made freely available and can be used following the links provided in the availability section. Even though the model can be further personalized employing immunological parameters and genetic information, we generated a few simulation scenarios for each patient based on the available data. Among these simulations, it was possible to find the scenarios that realistically matched the real clinical and MRI history. Moreover, for two patients, the simulator anticipated the timing of subsequent relapses, which occurred, suggesting that UISS may have the potential to assist MS specialists in predicting the course of the disease and the response to treatment.

## 1. Introduction

Multiple sclerosis (MS) is a heterogeneous disease resulting from the interaction among environmental and genetic factors with a dysregulated immune response. Substantial shreds of evidence indicate that MS is an autoimmune disorder in which an immune response is directed to myelin components, but neurodegenerative processes also occur and contribute to the buildup of neurological damage. The leading hypothesis on MS pathogenesis suggests that auto-reactive *T*-lymphocytes, exhibiting the Th1 and Th17 phenotype, become activated in the periphery and gain access to the CNS through a disrupted blood–brain barrier (BBB) [1]. Once in the brain, auto-reactive immune cells trigger inflammation by recruiting B lymphocytes, macrophages, and microglia. Tissue damage results from the release of inflammatory cytokines and other detrimental factors such as reactive oxygen species (ROS) and reactive nitrogen species (RNS) [2]. The mainstay of MS treatment is immunomodulating or immunosuppressive drugs and the therapeutic scenario has dramatically changed over the last two decades with the advent of several monoclonal antibodies and oral medications. About 20 drugs are now approved for the treatment of MS, but active substances are 13 as interferon 1 beta, glatiramer acetate, and dimethyl fumarate are marketed under several brand names. Over the last decade, disease-modifying drugs (DMDs) for MS have been classified by regulatory authorities as first-line, second-line, and third-line therapies, mostly based on their safety profiles. However, according to their evidence of efficacy, drugs for MS are now grouped by clinicians into moderate-efficacy DMDs and high-efficacy DMDs [3]. As of today, the assessment of therapeutic response can only result from clinical and magnetic resonance imaging (MRI) data and it requires months of careful monitoring to be reliable. In an era where therapies are oriented to personalization, tools to predict the evolution of MS and the response to treatments at the individual level are eagerly needed.

In recent years, the concept of early therapeutic intervention has become popular also in the field of MS. A substantial amount of data indicate that brain damage occurs from onset and proceeds relentlessly, even during clinically silent phases [4,5,6,7]. Furthermore, long-term studies show that patients who were treated earlier have best outcomes as far as relapse rate, disability accumulation, and brain volume loss [8]. The ideal target of MS treatment is now the achievement of no evidence of disease activity (NEDA), which encompasses the absence of clinical relapses, disability progression, and new MRI lesions [9]. The most common treatment strategy so far has been that of escalating from a moderate to a high-efficacy drug, if disease activity was still persistent or resurgent. However, for patients with highly active disease at the presentation, the use of high-efficacy therapies from the onset is recommended to prevent the accumulation of disability [10]. Furthermore, some of the high-efficacy DMDs such as alemtuzumab and cladribine have profound depleting effects on T and B lymphocytes. They may lead to a reshaping of the antigen-specific repertoire, reproducing to a minor extent the effects of autologous hematopoietic stem cell transplantation. These treatments are referred to as immune reconstitution therapies and represent the prototype agents of the induction strategy for MS [11].

Whatever the initial therapeutic approach is, a close clinical and MRI monitoring is warranted during the first year of treatment, as the recurrence of disease activity should prompt a switch to more efficacious drugs. For this purpose, MRI results are more sensitive than clinical parameters as new lesions tend to occur 5–10 times more frequently than relapses. However, performing MRI at short intervals is infeasible in daily practice. Besides, first-line drugs, although comparable for efficacy, exhibit different mechanisms of action (MoA) and individual response may be heterogeneous. Thus far, no laboratory test has been validated as a predictor of drug responsiveness or therapeutic failure in MS, and therefore the exploration for new and reliable tools to guide the treatment is still ongoing. Finally, it is largely believed that a personalized approach would be the most efficacious in the treatment of multifactorial diseases such as MS, but this clinical strategy is still far to be put into practice due to the high number of variables that need to be considered [12].

In the last years, different models focused on describing MS dynamics in general, and the relapsing–remitting multiple sclerosis (RRMS) type from a mechanistic point of view has been proposed. In this regard, an exhausting overview of computational modeling applied to MS has been published by Pappalardo et al. [13].

Vélez de Mendizábal et al. [14] presented a stochastic differential equations (SDE) model able to reproduce the typical oscillating behavior of RRMS qualitatively. However, such a model is quite simplistic, as it takes into account only a restricted number of entities involved in the development of the disease. In [12], the authors present an agent-based model developed with NetLogo software. The model is based on the same hypotheses of the previous one, but with a complete description of the involved entities, including some thoughts about the possible positive effects of vitamin D [15], and the role of the blood-brain barrier (BBB) [16]. However, both two approaches do not take into account any specific treatment.

Differently, in [17,18,19,20], a computational methodology based on Petri Net formalism has been proposed to analyze the RRMS behavior under daclizumab administration qualitatively.

Recently, Kannan and collaborators [21] developed a computational model that reproduces the main types of MS and accounts for several features of the disease progression. Kotelnikova and co-workers [22] presented a mathematical model that reproduced the different disease courses, despite the heterogeneity of genetic and environmental factors, supporting the hypothesis that common pathogenesis is at the origin of different MS subtypes, and this also applies when genetic and environmental factors are heterogeneous. Although the process by which constitutional (genetics, ethnicity, and gender) and environmental (viral, sun exposure and diet) elements would interact in MS pathogenesis is still far to be elucidated, data derived from the natural history of the disease and longitudinal MRI studies provide prognostic tools currently used in daily practice. It must be said that all the models described above limit their functions to a general qualitative description of the disease, and none of them is tailored explicitly with patient data to provide personalized simulations under the administration of a specific treatment. The model we present here is the only one that, to our best knowledge, is presently able to deliver an accurate description of the disease dynamics, including the effects of the most common treatments. Moreover, our model is designed to use specific patient data to provide personalized simulations.

A computational model of MS dynamics, like the one we are going to describe, could overcome the difficulties in handling a large quantity of information required to generate a disease profile at the individual level, helping to predict the course of the illness and to choose the best therapeutic option.

Based on these premises, we envisaged to build a computational model for reproducing and predicting the immune system dynamics in MS patients and we tuned the model with clinical and MRI features as a preliminary attempt to personalize the framework. The next step would be to incorporate immunological features (cytokines level, lymphocytes subpopulation counts, and HLA profile) to refine the in silico model and hence increase its predictive power. The final aim would be the generation of a baseline profile for each patient that could be translated into a computational model to assist clinicians in predicting the evolution of the disease and the response to different drugs.

## 2. Materials and Methods

### 2.1. The Computational Model

MS is a neuroinflammatory disease with an extremely complex immunopathology that involves practically all the immune system machinery. Moreover, as in the case of other autoimmune diseases, the building of a computational framework should also consider the role of immunoregulatory mechanisms. Having a methodology capable of describing the entities, along with their relationships, is needed. Ontology is an approach that formally defines and assigns the types, properties, and interrelationships among the entities within a particular domain. Building an ontology designed specifically for MS aims to give a substantial contribution to acquire a better understanding of the critical properties of the phenomenon, i.e., the system entities, how they are organized and their dynamics. Malhotra et al. [23] developed an exhaustive ontology from MS, using scientific literature and an expert review under the Protégé Web Ontology Language (OWL) environment [24]. Protégé is a complete editor for developing ontologies and it is released under an open-source license. It also allows, through the use of a reasoner engine, to check the consistency of the ontology. We used Protégé to build a complete ontology that contains: (i) the classes needed to describe both the immune system and specific disease related features [23]; (ii) the object properties related to the interactions and localization; (iii) entities (i.e., individuals) that are linked to each class and (iv) data properties that provide an extensive description of the interactions. Once the ontology description is completed, the next step is the choice of the best computational technique able to host all the components of interest.

Several immunological case studies [25] describe the use of unified modeling language (UML) exhaustively in representing and capturing the dynamics of individual model entities in system-level emergent properties. UML formalism could be a valuable complement in modeling complex systems, especially in immunology, and, in a future perspective, it could be useful to provide a metamodel combining both languages in the definition of a biological system. Both UML modeling and OWL ontology own strengths and weaknesses, similarities, and differences. It is worth mentioning that ontology could be used to represent interactions adequately. An example of this use is shown in Figure 1.

We used agent-based modeling (ABM) approach to build the simulation framework and taking into account our previous successful experiences in simulating the dynamics of the immune system in several pathologies. In particular, we adapted and extended our immune system simulation framework [16,26,27,28], named “Universal Immune Simulator Framework” (UISS), to model the dynamics of the immune system in MS along with the therapeutic effects of teriflunomide, fingolimod, IFN-β1a, ocrelizumab, and natalizumab. The advantages of this modeling methodology are well known: entities, as well as biological functions and interactions, can be described very closely to the biological reality; at the same time, this approach allows a mathematician or a computer scientist to describe the scenario using a logical and rational framework. Finally, the ABM approach allows flexibility and further extension and refinement of the model without significant additional effort.

The ABM is built over five main concepts. The identification concept includes entities with their functions and activities, playing both a role in the immune system dynamics and the evolution of MS. In particular, we considered all the relevant immune cells and molecules along with their specific properties.

The concept of localization is devoted to establishing the position in which the entities are located, especially during interactions. We considered two specific compartments, the brain (white matter) and the peripheral lymph nodes. Effector cells (B and T cells) are challenged with auto-antigens and consequently activated into the lymph node compartment while the inflammatory and immune system activity against the target disease takes place into the white matter compartment. The class named “treatments” includes all the details and the effects of the therapies we tested by the simulator. The biological scenario class identifies all the specific elements and interactions of MS we took into account. We describe in detail this concept later on. To characterize all the cellular and molecular activities, functions, and interactions, the computational framework was built as a polyclonal model, i.e., it can represent all the possible binding sites of B and T lymphocytes, major histocompatibility complexes of both class I and II, antigens (MHC-I and MHC-II), epitopes and peptides. The technique used is based on a binary string representation of the binding sites. Both molecular and cellular entities interact following procedures describing the recognition processes and the consequential events according to the immune system response. The model also implements processes encompassing phagocytosis, antigen presentation, cytokine and chemokines release, cell activation, cytotoxicity, and immunoglobulins secretion.

In our simulation framework, interactions among entities take place when they are in a range in which they can “sense” each other. The term “sense” means that two entities can interact if they are located at the same site of the structure developed for the simulation, i.e., in the same lattice point. We used a lattice that represents a specific volume of peripheral lymph nodes or blood. Entities interact following a specific stochastic rule that is a function of the Hamming distance between binary strings that, as told before, mimic the receptor binding. The presence of cytokines is also considered, and they can act by favoring or inhibiting the probability of the interaction.

The model takes into account innate and adaptive immunity (both cellular and humoral) and immune memory.

### 2.2. Introduction to the Simulation Framework

To model MS dynamics, we used the UISS simulator framework. It is based on the agent-based model paradigm that has been developed to represent the immune system response to general pathogens [12,29,30]. In computer science, agent-based models represent a simulation technique in which entities, conversely to differential equations systems, are followed individually. In this way, complex emergent behaviors can arise and lead to the prediction of non-coded dynamics.

The entities, namely “agents”, are usually placed on simulation space (i.e., a lattice), and in each lattice point, we can have many entities. Such agents can be heterogeneous, can have internal properties (i.e., lifetime, internal state, and energy) and can act (i.e., move, interact with other agents in their neighborhood, modify their internal state or die) individually or as a result with the interaction with other agents. ABM has several advantages. By definition, they can be stochastic and include both delays and a spatial description. Furthermore, they permit to better describe the biological aspects and behaviors of the involved entities. As a consequence of that, the accuracy in the description is usually limited by the biological knowledge rather than the modeling methodology. Nonlinear behaviors, as well as the capacity to add further complexity and biological knowledge, do not represent a problem in solving the model. Such methods are also intrinsically numerically stable as integer numbers represented most of the involved variables, so very few complex floating-point operations are required.

UISS is entirely written in ANSI C-99 standard programming language, allowing us to obtain an architecture-independent platform. The modeling approach used by UISS is the same used by Celada and Seiden [31].

In UISS, we considered both cellular and molecular entities. Usually, cellular entities are followed individually and are modeled as singular agents. Molecules are instead considered by their concentration per lattice-site. Cell agents have different properties, such as position, half-life, and an internal state from a particular set of suitable states. Their dynamics are realized through state changes. A state change takes place when a cell agent interacts with another agent, like a cell, a molecule, or both of them. The most relevant cells belonging to the immune system have been considered, including B lymphocytes, helper, cytotoxic and regulatory T lymphocytes, and natural killer cells. Monocytes are represented as well and we take care of macrophages and dendritic cells. Some entities, like B and T cells, also have specific receptors for modeling specificity, like their real counterparts. For what concerns molecules, the model distinguishes between simple small molecules like interleukins or signaling molecules in general and more complex molecules like immunoglobulins and antigens, for which we need to represent the specificity.

At the same level of entities, immune system activities are represented. They include both interactions and functions. Functions refer to the central immune system tasks. In particular, UISS takes care of the diversity of specific elements, major histocompatibility classes restriction, clonal selection by antigen affinity, thymus education of T cells, antigen processing and presentation (both the cytosolic and endocytic pathways are implemented), cell–cell cooperation, homeostasis of cells created by the bone marrow, hypermutation of antibodies, cellular and humoral response and immune memory.

In UISS, as in most ABM approaches, time is discrete. It means that all system activities are followed and measured using equally spaced time intervals. At each time interval, all the immune system activities, such as the interaction and the diffusion processes, hold.

An interaction between two entities is a complex action, which eventually ends with a state change of one or both entities. The entities must be “near enough” to interact. More specifically, physical proximity is modeled through the concept of the lattice-site. All interactions among cells and molecules take place within a lattice-site in a single time step, so that there is no correlation between entities residing on different sites at a fixed time. In UISS, the simulation space can be either represented by a 2D *L × L* hexagonal lattice (six neighbors) or by a 3D as an *L × L × L* cubic lattice, with periodic boundary conditions or using rigid walls on the edges, according to the problem we are dealing with. All entities are allowed to move with a uniform probability between neighboring lattices in the grid with an equal diffusion coefficient (Brownian motion). This simulation space is used to represent, more from a biological point of view rather than a physical point of view, three anatomical compartments: the thymus, the bone marrow and a portion of a generic secondary organ.

Interactions can be seen as Bernoulli events, so each interaction has a given probability *p* to happen. Interactions can be classified as aspecific or specific interactions. Aspecific reactions are those that refer to the use of aspecific receptors. For example, if we take into account Toll-like receptors (TLRs), we know that they recognize with low specificity pathogen-associated molecular patterns (PAMPs) expressed by pathogens.

These will not explicitly be modeled in UISS, but instead, a fixed probability *p’* is used for all the interactions that involve the same couple TLR-PAMP. Specific reactions are those who involve cells coming from adaptive immunity and that is equipped with specific receptors. Specific interactions need a recognition phase between the two entities; in this case, the probability *p* of interaction depends upon the result of the recognition phase, in which the affinity between the involved receptors plays a major role.

UISS represents receptors and ligands as binary strings and uses a string-matching rule to model affinity. This smart idea introduced by Farmer and Packard [32] represents a simple way to mimic the typical molecular complementarity mechanism between receptors. While this may seem a rough approximation of the real biological event, millions of recognitions can be executed efficiently, allowing the study of large-scale properties of the immune system. Furthermore, models based on this approach produced accurate results when benchmarked to experiment, suggesting that the abstraction captures essential features of receptor/ligand binding and does not represent a limiting factor for the study of many biological scenarios [27,33]. The binding rule of the string-matching procedure mimics as well as the complementarity mechanism between two receptors by using the Hamming distance. This distance measures the number of mismatching bits between two strings.

As a consequence of that, repertoires are represented in the model as sets of strings, and the set of lymphocytes receptors is represented by bit-strings of length *h* that forms the so-called “shape space”. A clonal set of cells is characterized by the same clonotypic receptor, i.e., by the same bit-string of length *l,* the potential repertoire of receptors scales as 2^l^.

UISS can then be defined as a bit-string polyclonal lattice method. Bit-string refers to the fact that molecules and specificity among molecules are represented, and polyclonal identifies the capability to have more clones of different specificity of lymphocytes. Finally, lattice means that a discrete lattice is used to represent the space.

Hematopoiesis and thymus selection represent two of the most critical processes that regulate the immune system functions. Hematopoiesis is a biological process that describes, for example, the formation of blood cells derived from hematopoietic stem cells. In UISS, such a process is used to describe the production of B and T lymphocytes in the “bone marrow compartment”. Moreover, a further selection of T helper cells (TH) and cytotoxic T cells hold in the “thymus compartment” out by thymus selection.

Hematopoiesis is modeled as an Ornstein–Uhlenbeck mean-reverting process to keep the system in a metastable state in the absence of perturbations (cell homeostasis). The following differential equation describes this process:$$\frac{dx_{t}}{dt}\left(t\right)=\frac{ln2}{\tau_{i}}\left(x_{i}\left(0\right)-x_{i}\left(t\right)\right)+\frac{d}{dt}\left(t\right)$$

Here, $\overset{¯}{x}$ is the typical mean number of entities, x(t) the number of entities at the time, t,
η is the speed of the process to revert to the mean value and σdε(t) is random stochastic noise. At each time step, the formula is applied to calculate the number of cells that should be added (or removed) from the simulation.

Note that in the presence of immune response, the number of lymphocytes is always higher than the initial value. It means that $dx_{i}/dt<0$, so more cells will be removed than in the absence of an antigen. On the other hand, cell clones specific for the antigen, which can become antigen-specific memory cells, grow. The combination of these two actions causes the shift of the immune repertoire towards specificities that recognize the antigen.

Thymus selection is instead used to guarantee that in UISS a repertoire of MHC-restricted self-tolerant T cells is available. The process uses a selective procedure to mimic the real biological events that happen during T cell maturation in the thymus. This procedure is composed of two sub-sequential stochastic procedures: a first positive selection and a second negative selection. In the first phase, T cells with a receptor that has low affinity to the MHC molecules taken alone (class I for cytotoxic T cell (TC) and class II for TH) are removed as they are not useful.

Those cells that surpass the first phase are exposed to the second selection phase, in which T cell receptors are compared with an MHC-self-peptide complex. If a high affinity between the two strings is found, such T cells are eliminated to avoid autoreactivity.

Of course, receptors, MHC molecules and (self) peptides are always represented as binary strings, and the affinity is always calculated as described before. The self-antigen is defined by specifying in the input datafile of UISS, a subset of binary strings considered as self-peptides in the set $\left\{0,2^{l}-1\right\}$, where *l* is the chosen binary-string length.

We note here that a T cell receptor is positively selected if it matches at least one MHC molecule with an affinity greater or equal to $l/2-m_{0}$, where $m_{0}$ is the minimal number of mismatching bits.

Many other biological processes that are part of the immune system machinery are implemented. These include Hayflick limit on cell duplication, management of immune memory, hyper-mutation of antibodies, bystander effect, isotype switching, anergy, antigen digestion and presentation, and B cell receptor mutation.

As already described, entities may have several internal states depending on their type. We recall here the most common internal states, sketching some typical scenarios that bring to their appearing.
Newborn cells are introduced in the resting state.A cell becomes active when it is stimulated through an interaction with another entity. For example, TH cell activation occurs with the interaction with an antigen if TCR binds the antigen.Anergic state. In this state, the cell does not interact. This state applies, for example, to B, TH, and TC cells.Status intern applies only to antigen-presenting cells (APC). When an APC (i.e., an M, DC, or B cells) encounters the antigen, it may be directed recognized by membrane-bound receptors, for example, those on the surface of a naive B cell. Unlike B cells, T cells do not recognize antigens directly. They “see” antigen as peptides only in association with the host surface MHC molecules. Since MHC molecules can only bind peptide molecules of 7–15 amino acids long, T cells only recognize their specific antigen in the form of short peptides. The antigen presenting cells such as dendritic cells and B-cells take up antigen and partially degrade it into peptides, which then occupy the antigen-presenting groove in MHC class I and MHC class II molecules.When cells present peptides via MHC molecules, their status change to PresI (MHC class I) or PresII (MHC class II).Status duplicating (applies to TH, TC, and B cells) is achieved when a cell has been activated and stimulated to start the clonal division.BoundToAb status applies to specific cells (for example, pathogen-infected cells). This state represents the fact that a cell has been recognized by an antibody (Ab). Cells in this status may die by the action of Ab-complement or through natural killer cells.Status infected applies to virus target cells. It identifies viral penetration via permeabilization of the host cell membrane when it begins local replication and spreads. Each cell can be in different internal states and all cells are tracked individually throughout an experimental run. This status can be actively or silently infected, according to the fact that the virus is replicating or not.

Interactions represent the most important source of entities state changing and drive the evolution of the system. We can have different types of interactions, which include “molecule–molecule” interactions, “cell–molecule” interactions and “cell–cell” interactions, or a combination of them. A list of some of the most important interactions (far from being complete) of UISS is finally sketched as follows.
Epithelial cell-pathogen. If an epithelial cell encounters, for example, a specific virus or intracellular bacteria, the cell is infected by the pathogen (status infected), then the cell may also present its infected status (PresI) to the immune system cells. Epithelial cells are reported here as an example, hence we can have different cells representing the different targets of different pathogens.IG_Ag. If a soluble immunoglobulin (IG) encounters its specific antigen (Ag), the IG binds to Ag and forms an immunocomplex (a macrophage can capture that). The binding probability is, as already said, a function of the Hamming distance of the two entities.B_Antigen. If a naive B lymphocyte expresses at the cell surface a membrane IG, which is specific for the native antigen (calculated according to the Hamming distance between the two strings), the B lymphocyte internalizes the membrane IG and the bound Ag (state intern). It then processes the IG–Ag complex into peptides, which are then presented by MHC-II (status PresII) at the B lymphocyte surface. We recall here that the binding probability is a function of the Hamming distance of the B receptor and the peptide (specific interaction). The B cell is now an APC (specialized antigen-presenting cell).M_Antigen. If a macrophage encounters a native antigen or an immunocomplex, the macrophage internalizes the antigen or the immunocomplex (state intern). Then it processes it into peptides, which are then presented by MHC-II (status PresII) at the macrophage cell surface. The M is now an APC (specialized antigen-presenting cell).DC_Antigen. If a naive dendritic cell encounters a native antigen or an immunocomplex, the dendritic cell internalizes the antigen or the immunocomplex (state intern), and then it processes it into peptides, which are then presented by MHC-II (status PresII) at the dendritic cell surface. The DC cell is now a specialized antigen-presenting cell (APC).TH_B. If a resting T helper lymphocyte (TH) encounters the B lymphocyte that is presenting a given peptide-MHC-II complex (status PresII), the TH cell becomes an activated T helper lymphocyte (status active) that helps the B cell to differentiate into the plasma cell or memory cell. At the molecular level, the interaction holds if the T cell receptor (TR) at the surface of the Th cells binds specifically to the peptide-MHC-II complex (specific interaction). Then active Th proliferates and secretes interleukin 2 (IL2). At the same time, B lymphocyte proliferates and differentiates into a plasma cell (that secretes IG) or into a memory cell (with IG at its surface).TH-M and TH-DC. As already shown in the TH_B interaction. If a resting TH encounters a macrophage or dendritic cell in the PresII state, TH becomes activated and secretes interleukins that activate other cells of the immune response (NK, mast cells, cytotoxic T lymphocytes, and others).TC–epithelial cell. If cytotoxic resting T lymphocyte (TC) encounters an epithelial cell that is infected (and thus also PresI status), for example, by a virus or intracellular bacteria, the cytotoxic T cell becomes, in the presence of IL2, an active cytotoxic T lymphocyte that kills the other cell. At the molecular level, the T cell receptor (TR) at the surface of a resting cytotoxic T lymphocyte binds specifically to the peptide-MHC-I complex at the surface of the cell. It must be noted that such interaction with TC cells may also arise for other infected cells such as, for example, M infected by tuberculosis.IG–epithelial cell (IG–bacteria, IG–virus). If an immunoglobulin IG encounters an infected epithelial cell that is presenting the antigen at its cell surface, and the soluble IG recognizes specifically the antigen, the opsonized cell (cell with bound IG on its surface) may be killed by complement-dependent cytotoxicity (CDC) or by antibody-dependent cell cytotoxicity (ADCC) mechanisms. At the molecular level, the first interaction is the recognition by the IG of the antigen expressed at the surface of the bacteria. A similar specific interaction with IG may also arise in other scenarios that include binding of IG to cancer cells, infected cells, viruses, bacteria, and others.

Before starting the simulation, an initialization phase is executed. In this phase, the lattice is populated with the required number of entities. Then, the simulation is carried out for a given number of time-steps. At each time-step, both the interaction-driven processes and the non-interaction driven processes (i.e., movement or internal processes) are executed. It must be noted that, ideally, all processes within the time step should happen synchronously. As this does not represent a problem for movement or for internal dynamics that can be easily reproduced as serialized processes, it may be a possible source of bias for the interaction dynamics.

A total massive parallel execution is challenging to achieve in actual computers, especially when the number of entities is very high. To this end, to minimize any bias, for each lattice-site a different random interaction scheme is generated both considering a random order of the interaction rules and a random order in the list of agents that may interact within the same rule. For a given rule that refers to two entity types A and B, every entity of type A is compared with all the entities of type B within the same site until a successful interaction occurs. Then, the next type A entity is compared again with all entities of type B. Next interaction rule is finally executed as soon as the entities have had their chance to interact.

In UISS a set of core features is used to reproduce the standard immune system machinery and its response against a generic pathogen. This core that is composed of entities, processes, and interactions, does not usually change unless novelties in the biological knowledge are discovered and assessed.

The set of basic features is extended to simulate new pathologies, from time to time, with a set of specific features necessary to reproduce the missing involved biological mechanisms. While such extensions may change from one pathology to another, the core is kept.

### 2.3. Extension of the Simulation Framework to Include MS Pathogenesis and Related Treatments

MS affects the brain and spinal cord [2,34], but we restricted to the brain oligodendrocytes (mostly present in the white matter region), the main target to measure demyelinating hits throughout the disease. In particular, myelin basic protein (MBP) represents the oligodendrocyte-associated antigen that we used to simulate the self-antigen recognition by the immune competent cells. The complete list of entities specifically used and/or added for modeling MS is reported in Table 1.

The model takes into consideration a 2D domain physical space: although this limits the representation space, all the processes and interactions are simulated between two main compartments. Lymph nodes represent the first one while the second one is represented by a slice of white matter populated by oligodendrocytes. This representation results in a satisfactory approximation and allows the simulator to run smoothly.

To set the simulation space, we used the same approach followed in [35]. To simulate inflammation and the resulting chemoattraction dynamics, we also implemented into the simulation framework the chemotaxis feature. Chemotaxis is a biological phenomenon in which some cells, such as immune system cells, direct their movements according to specific chemical signals. Chemotaxis is a very complex process not yet fully understood. It involves many factors, such as short and long-range interactions. To our best knowledge, there is no model able to represent this phenomenon completely.

In a first attempt to mimic short-range chemotaxis effects, higher probabilities of being chosen are given to sites containing chemokines. Let $x_{0}$ be a lattice-site and $x_{i},i=1...6$ be its neighborhood. For every $x_{i},i=1\ldots 6$, the probability pi is defined as:(1)$$p_{i}=\{\begin{array}{c} \frac{C_{i}}{\sum_{j=0}^{6}C_{j}},\sum_{j=0}^{6}C_{j}>0\\ \frac{1}{7},\sum_{j=0}^{6}C_{j}=0 \end{array}$$
where $C_{i}$ represents the number of chemokines in the lattice site $x_{i}$. Once a new site $x_{i}$ has been chosen for movement, the entity moves towards $x_{i}$. with probability *p* calculated as the following:(2)$$p=1-min\left(\left(\frac{\sum_{i}^{n}E_{i}}{M_{cells}}.K\right),1\right)$$
where $\sum_{i}^{n}E_{i}$ denotes the total number of entities $E_{I}$ in the lattice-site x, K is a constant ∈(0,1) and $M_{cells}$ is the maximum number of entities a lattice site can usually contain. The use of K is justified by the fact that in non-safe conditions (i.e., in the presence of inflammatory processes), biologists observed the infiltration of immune system cells such as macrophages in situ. Thus, using K<1 it is possible to have some chance for immune system cells to move into sites that are already full. In the absence of chemokines, cells move through a random walk.

The simulator takes care of the main interactions that happen during an autoimmune response. Besides the interactions that we already have in the simulation framework, we added specific interactions that are needed to simulate the inflammatory and autoimmune response against oligodendrocytes. We included the following interactions.

*Resting dendritic cell and oligodendrocyte interaction*. If a dendritic cell (DC) encounters an oligodendrocyte through a cross-presentation mechanism, DC captures and cross-present antigens from oligodendrocyte as peptide/MHC-I complex. In this case, since an oligodendrocyte is a self cell, B7 proteins are not activated in DC.

*Antigen-presenting in peptide/MHC-I complex dendritic cell with B7 proteins not activated and cytotoxic T cell interaction*. If in a lymph node, a DC that is presenting on its surface peptide/MHC-I complex (APC) with B7 proteins not activated, encounters a resting cytotoxic T lymphocyte, in the context of defective peripheral immune tolerance, cytotoxic T cell (auto-reactive clone) becomes activated. This scenario simulates what happens in individuals where genetic predisposition combined with environmental stresses can lead to a breakdown in peripheral tolerance leading to autoimmunity [36].

*Oligodendrocyte with IgG interaction*. If a self-reactive IgG recognizes the antigen expressed on the membrane of the oligodendrocyte, their binding leads to opsonization. The opsonized cell (cell with bound IG on its surface) may be killed by complement-dependent cytotoxicity (CDC) or by antibody-dependent cell cytotoxicity (ADCC).

*An activated macrophage with IgG coated oligodendrocyte interaction*. If a macrophage encounters an opsonized oligodendrocyte, the macrophage kills the oligodendrocyte and releases chemokine factors in the site.

*Activated cytotoxic T cell with oligodendrocyte interaction*. Into the white matter, if a cytotoxic T lymphocyte (CD8) encounters an oligodendrocyte, CD8 T cell kills the oligodendrocyte. Oligodendrocyte killing promotes the local secretion of chemokine factors. At the molecular level, the CD8 receptors recognize the MBP/MHC-I complex shown at the oligodendrocyte surface.

All of these interactions were implemented into the ontology. Figure 1 describes one of the most common autoimmune responses against oligodendrocyte, i.e., a CD8 activated T cell that kills the oligodendrocyte.

Although the etiology of MS is still mysterious, environmental, genetic, infectious and immunological factors contribute to MS pathogenesis and may be also be conceived as risk factors [1]. In our simulation framework, we considered all of them with one or more representatives in each class of risk factors. In particular, we included the Epstein-Barr Virus (EBV) infectious agent that is believed necessary to the genesis of MS [37]. We simulated the mechanism of molecular mimicry that involves reactivity of T and B cells with either peptides or antigenic determinants shared by infectious and self-antigens (MBP in this simulation, which is a potential autoantigen). Genetic predisposition was considered by simulating, into the binary string that implements human leukocyte antigen (HLA) genetic composition, those polymorphisms that render patients more prone to activate an autoimmune response against self-antigens expressed on the surface of an oligodendrocyte (ODC; e.g., MBP). For the specific immune system class, we considered a malfunction of regulatory mechanisms: this presently involves both central and peripheral immune tolerance malfunctioning. We simulated central malfunction through an impaired thymus selection process that produces a more significant number of auto-reactive T cells. Peripheral tolerance malfunction was implemented through an increased probability for an auto-reactive clone (both T and B) to be activated even in the absence of B7 proteins expressed on the surface of APCs. It prevented both induced anergy of autoreactive clones and a sufficient release of anti-inflammatory cytokines like IL-10, IL-4, and TGFB. Environmental factors were represented by low vitamin D levels and cigarette smoking as both were demonstrated to play a detrimental role in the immune system regulatory mechanism and MS. This risk factor matrix allowed us to obtain a good framework that can simulate the baseline conditions for a virtual patient developing MS. Another goal of our work is to study the evolution of relapsing MS in patients under licensed treatment such as teriflunomide, fingolimod, IFN-β1a, ocrelizumab or natalizumab. Consequently, we needed to implement their mechanism of action inside the simulation framework.

To include the effects of the IFN-β1a drug into the model, we searched for literature containing a summary of biological activity and dynamics of IFN-β1a. Laboratory and clinical studies [38] have shown that IFN-β1a modulates inflammation underlying MS activity, acting on a variety of processes and molecular mediators within the immune system. In particular, we inserted: (i) IFN-β1a effect on the cytokine production, favoring the switch from a Th1 (pro-inflammatory) to a Th2 phenotype (anti-inflammatory) [39]; (ii) inhibition of T-cell activation and reduced expression of MHC-II molecules, which in turn reduces self-antigen presentation [40] and (iii) enhancements of CD4+ regulatory T cells activity [41]. For modeling purposes, we considered IFN-β1a at the dosage of 44 mcg three times a week.

Teriflunomide acts through a specific non-competitive reversible inhibition of the enzyme dihydroorotate dehydrogenase (DHODH), localized within mitochondria organelle and highly expressed in proliferating lymphocytes [42]. This drug inhibits cell growth and the division of proliferating T and B cells. To build the model of this mechanism of action, we extracted the topography of all the entities and the relative biochemical reactions from KEGG (Kyoto Encyclopedia of Genes and Genomes) PATHWAY Database [43]. With all these data ready, we developed a model network made of 29 different species involved in 13 enzymatic reactions. The model was realized through a complex pathway simulator (COPASI), a software able to simulate and examine the dynamics of biological pathways [44]. The model accepts as an input the dosage of teriflunomide and provides, as an output, the probability to reduce T and B cell proliferation to the main simulation framework. For modeling purposes, we considered teriflunomide at the dosage of 14 mg once a day.

Fingolimod is an oral prodrug that acts by inhibiting lymphocyte egress from lymph nodes, especially interleukin-17-producing T cells [45]. We simulated the net effect of the fingolimod by preventing T cells migration into the compartment of white matter and by inhibiting IL-17 producing cells (Th-17). For modeling purposes, we considered fingolimod at the dosage of 0.5 mg once a day.

Natalizumab is a humanized monoclonal antibody that selectively binds to the α4-subunit of α4β1- and α4β7-integrins expressed on the surface of human leukocytes [46]. We simplified the mechanism of action, taking into account the final effect that can be reproduced as a reduction of leukocyte migration in the white matter compartment [47,48]. For modeling purposes, we considered natalizumab at the dosage of 300 mg every four weeks. Ocrelizumab is a humanized monoclonal antibody directed against CD20 B cells [49] and to represent its activity we exploited the processes already included in the computational framework that was both antibody–cell interaction and natural killer–Ab-opsonized cell interaction. In other words, when ocrelizumab is injected, it binds to B cells, which become specific targets for natural killer cells. For the modeling purposes, we considered ocrelizumab at the dosage of 600 mg every six months.

### 2.4. Simulation of Real Patients

The main target of this study is to have a simulation framework able to predict the dynamics of relapsing–remitting MS patients when treated with specific drugs. After the written approval of our local ethics committee (Comitato Etico dell’Azienda Ospedaliera “Garibaldi”), the six patients gave their informed consent to participate to this study. The number of Ethic Committee approval is 2/2019/CECT2. In [50], the authors present a multivariable survival and generalized linear models to predict individual treatment response for MS. However, they use a statistical approach that is very useful to identify patterns in the prevalent multiple sclerosis population, whose predictive value exceeds that of the isolated individual variables. We used a different approach that was based on a mechanistic model. The aim is to have a detailed analysis of the immune system behavior that could predict the outcome of treatment in a specific MS patient. We limited the analysis of drugs to IFN-β1a, fingolimod, and teriflunomide (first-line drugs) and natalizumab and ocrelizumab (second-line drugs). The simulator offers various strategies to reproduce a real patient in silico, hence to personalize the simulation. The optimal case would be a patient at disease onset with specific analyses that can provide data about: (i) cytokine levels, i.e., IL-2, IL-10, IL-12, IL-17, IFN-G, and TGFB; (ii) the HLA genotype; (iii) MRI lesion load; (iv) lymphocyte subpopulation data; (v) oligoclonal bands status; (vi) vitamin D levels; (vii) history of infections; (viii) gender and (xix) ethnicity. To obtain all the data mentioned above in real patients, we should conduct a prospective and systematic study, but, at this stage, this would not be feasible without preliminary evidence that our simulation framework could be useful for the purpose. After the approval of our local ethics committee, we selected six patients who gave their informed consent. For each subject, we obtained the age at MS onset, baseline MRI lesion load, oligoclonal bands status, and the administered treatment. These features are widely accepted as prognostic parameters of MS evolution [51]. For the modeling aims, we associated the lesion load at the onset with the probability for an oligodendrocyte to die by apoptosis, hence releasing in situ autoantigens that can then be captured by APC (dendritic cells and macrophages). The greater the lesion load at the onset, the higher is the probability for an oligodendrocyte to die by apoptosis. The presence of oligoclonal bands is a biomarker that is associated with a more aggressive prognosis of MS [52,53]. As the oligoclonal bands are made of immunoglobulins of class G and/or M, they can be translated, in the simulation model, into an augmented autoimmune response in patients suffering from MS. Then we associated the presence of oligoclonal bands with an impaired function of the regulatory system in the simulated patient. Finally, we considered the age as an indicator of generic immune system reactivity. The lower is the age at the onset, the more active is the general immune response against auto-antigens. Table 2 shows a summary of the clinical data obtained for each of the selected individuals.

Another aspect that we took into account was the pregnancy condition of some female patients. Several findings report that pregnancy ameliorates MS and other cell-mediated autoimmune conditions like psoriasis and rheumatoid arthritis through a temporary status of enhanced immunomodulation [54,55]. In this case, during the pregnancy period, we simulated an increased activity of the central and peripheral mechanisms of immune tolerance, leading to a lesser activation of auto-reactive T and B clones directed against ODC.

## 3. Results

The computational framework was able to reproduce the biological variability of the immune system repertoire. Setting an initial parameter that identifies the random seed of the simulation, the binary strings that are related to MHC-I and II were randomly chosen. Consequently, the initial immune system repertoire was shaped randomly selecting lymphocytes receptors that undergo, in a second step, to thymus selection (for T lymphocytes) and bone marrow maturation (for B cells and the other immune system cells). In other words, varying the initial random seed, the simulator was able to reproduce different immune repertoires for an identified patient. This feature is really critical for almost all the simulations that involve the immune system dynamics.

### 3.1. Prediction Robustness

Based on Rotstein and Montalban’s last pieces of evidence [56], we considered some of the major predictor factors established for poor prognosis in multiple sclerosis. They dealt with demographic and environmental factors, clinical factors, MRI observations, and biomarkers. We used some of them, i.e., age, vitamin D levels, smoking, number of lesions, lesions volume and IgG and IgM oligoclonal bands in cerebrospinal fluid (CSF), as simulator input parameters to verify poor prognostic factor evidence. To achieve a proof of model correctness and robustness, and to reproduce biological diversity, we generated a sample set of 100 virtual subjects-specific models, from which we extracted 60 relevant behaviors of MS patients, classified in different levels of disease severity: low (20), medium (20) and high (20). The disease severity was measured in terms of the loss of ODC.

In Figure 2, we show both the mean behavior (green line) and the ±SD (shaded region) of the ODC dynamics respectively for those patients that show respectively a low, medium, and high level of ODC loss. Each cluster of patients is characterized by different values of age, vitamin D, smoking, number of lesions, lesions volume and IgG and IgM oligoclonal bands in CSF in a way that represents the corresponding prognostic severity scores.

Results in Figure 2 reflect the influence of the factors identified by Rotstein and Montalban in the prognosis of MS. Panel (a) shows a light number of relapses (measured as ODC loss): the simulated patients, in this case, were affected by normal levels of vitamin D, light levels of lesions in terms of numerosity and volume, absence of oligoclonal bands, middle age and a negligible habit in smoking. Panel (b) describes a medium number of relapses (measured as ODC loss): the simulated patients, in this case, were affected by low levels of vitamin D, medium levels of lesions in terms of numerosity and volume, presence of oligoclonal bands, middle age and a not negligible habit in smoking. Finally, panel (c) depicts a high number of relapses (measured as ODC loss): the simulated patients, in this case, were affected by very low levels of vitamin D, high levels of lesions in terms of numerosity and volume, presence of oligoclonal bands, older age and a not negligible habit in smoking.

### 3.2. Real Patients’ Predictions

We designed real patients’ simulations using the following procedure. For each of the six patients, we set the simulator parameters based on the clinical, laboratory and MRI features of the patients, i.e., age at the onset, lesion load at the onset, and oligoclonal (OB) status. Given the wide variability of predictable MS dynamics stemming from only three baseline features, for each virtual patient, we ran the simulator using multiple different immune system repertoires randomly generated. We verified that 10 were the minimum number of simulations that allowed us to obtain at least one model that could reasonably fit the available clinical data. These findings demonstrated that the simulation framework was potentially able to reproduce the real individual dynamics of MS. Figure 3 (panels (a)–(f)) shows the simulations representing dynamics of ODC and other variables related to immune system, which best fitted the clinical data obtained for each patient.

We simulated three cubic millimeters of white matter with ODC content explained before, i.e., 50 × 10^3^ ODC per cubic millimeter. We set as an MRI detectable event a lesion that involves at least 50 × 10^3^ ODC, approximating a uniform distribution of ODCs in the white matter tissue. If the ODC loss is greater or equal to 50 × 10^3^, this quantitatively means that one or more lesions appeared and are MRI detectable. We were, at this stage, unable to have a better quantitative measure. We are working to have another quantitative method to put in the relationship with what the model measures in terms of ODC loss with real biological data. One of these may be represented by DNA methylation of the myelin oligodendrocyte glycoprotein gene in the blood [57]. Smaller ODC losses (undetectable by MRI) were classified as inflammatory biological activity and were considered useful for the analysis of treatment response. The data from Table 2 and Figure 3 show that the predicted ODC dynamics were in good agreement with the MRI changes in lesion load. Predictions of potential adverse reactions were obtained for each patient, increasing the simulation time for one year. Furthermore, for patient id 2789 and patient id 2070, the computational framework forecasted two events of ODC injury, which were confirmed afterward by clinical or MRI follow-up. It confirms that the simulation infrastructure was also able to correctly predict future relapse events due to a therapy that was potentially not effective for that patient.

In Figure 4 and Figure 5 we reported further simulation plots to show both the immune system dynamics and oligodendrocytes dynamics while comparing an independent virtual healthy control case with a MS patient. The simulation time was 4.5 years for both the two scenarios. It is worth highlighting that to reproduce the typical dynamics in a relapsing-remitting MS virtual patient, we set two specific parameters in the computational framework that control the regulatory (both central and peripheral) system.

In each plot, it is depicted the dynamics over time of oligodendrocyte (ODC), Active T helper cells (Active Th-1), Resting T helper cells (Resting Th-1), Active regulatory T cells (Active Treg), Resting regulatory T cells (Resting Treg), Active cytotoxic T cells (Active Tc), Resting cytotoxic T cells (Resting Tc) populations, and inflammatory cytokines Interleukin-2 (IL-2) and Interferon-gamma (IFN-g). In Figure 4, panel (a), the variations in the levels of oligodendrocytes are relatively smaller in healthy controls than those of the diseased cases (Figure 5, panel (a)), showing that no immune system activity is in progress; in the simulated MS patient four relapsing events occurred, showing ODC loss and then ODC recover (Figure 5, panel (a)). TH cell population levels show, in the MS patient, an increase of the primed helper T cells type 1 (Figure 5, panel (b)) in respect to the healthy patient (Figure 4, panel (b)). These helper T cells type 1 drive the consequent auto-reactive cytotoxic activity against ODC, led by CD8 T cells (Figure 5, panel (f)). Such a strong cytotoxic activity is not present in the healthy patient (Figure 4, panel (f)). Regulatory T cells activation was noticeably increased in the healthy control (Figure 4, panel (d)) as it regulated and suppressed the auto-reactive immune response, while in the MS simulated patient this activity was sensibly reduced (Figure 5, panel (d)). The inflammatory interleukin 12 and interferon gamma cytokines level were strongly increased in the MS patient (Figure 5, panels (h) and (i)) compared with the healthy ones (Figure 4, panels (h) and (i)), showing an inflammatory condition that promotes consequent immune system activation. 

There are several models in the literature [12,14,16,21,22] that deal with MS modeling. However, they do not take into consideration specific immune system details and, more importantly, were not used in clinical settings to predict the outcome of MS therapies. Of course, our attempt was far from being complete, but it is potentially ready to be verified in a systematic clinical study that we hope to set in due course.

The complete simulation framework is made freely available following the link https://combine.dmi.unict.it. Visitors have to sign up to launch the simulation framework. The interface to the simulator was developed using Flask micro server environment and python programming language.

## 4. Conclusions

We presented a computational framework based on the ABM paradigm that, to our best knowledge, can deliver an accurate description of the disease dynamics, including the effects of the most common treatments. It improves the current state of the art, also making use of specific patient data to provide personalized simulations, mirroring the course of the relapsing-remitting phase of MS based on individual clinical, laboratory, and MRI features. Moreover, the simulator was able to predict the effects of the most used drugs against MS.

Currently, patients suffering from the first relapse of the disease (even with minor symptoms) are evaluated by a neurologist, who performs a brain MRI that confirms the presence of typical MS lesions. This diagnostic test hence represents the gold standard procedure used to diagnose MS and to track the disease’s progression, also due to its non-invasiveness and sensitivity way of imaging the brain. Hence, the costs of the model do not impact at all during the diagnosis process as MRI scanning is a mandatory diagnostic requirement; instead, the computational framework presented in this work could drastically reduce the costs deriving through a wrong therapy selection.

The final goal of our effort is to build a comprehensive profile, including also genetics, immunological, and environmental data, so to reliably predict the real dynamics of MS at the patient level and inform the therapeutic choice at early stages. The use of in silico models to simulate the clinical course of the disease and response to therapeutics has been recently acknowledged and recommended by regulatory agencies. Our model is a preliminary attempt to reproduce and predict the clinical and therapeutic dynamics at an individual level and needs to be validated on a larger population and to be integrated with supplementary information to better shape patient profiles. Once refined, this simulator may reveal a helpful tool to assist MS specialists in predicting the course of the disease at early stages and selecting the best treatment strategy, circumventing the need for a trial and error procedure.

## Figures and Tables

**Figure 1 cells-09-00586-f001:**
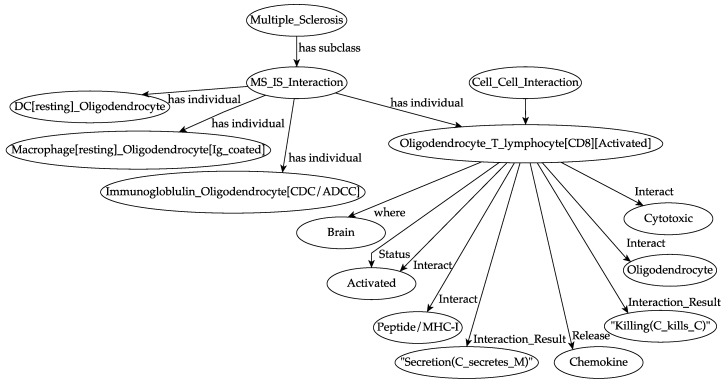
An overview of the ontology of a specific autoimmune interaction during multiple sclerosis (MS) dynamics. A previously activated cytotoxic T lymphocyte encounters its target, i.e., an oligodendrocyte, in the white matter portion of the brain. It recognizes, at the molecular level, the peptide/major histocompatibility complex (MHC-I) that is exposed on the surface of the oligodendrocyte. For each entity, the localization (i.e., the biological compartment in which the entities are present, in this specific case the brain) and the status (i.e., the differentiation states that an entity can own, in this specific case “activated”, that means primed) are defined. The result of the interaction is the killing of the oligodendrocyte by the cytotoxic T lymphocyte and the local release of chemokine factors. Specifically, “C kills C” means that a cell, for example, a CD8 T lymphocyte, kills a peptide/MHC-1 presenting oligodendrocyte. “C secretes M” means that a cell, in this case, an oligodendrocyte, secretes specific molecules (i.e., chemokines) that attract further activated effector immune cells.

**Figure 2 cells-09-00586-f002:**
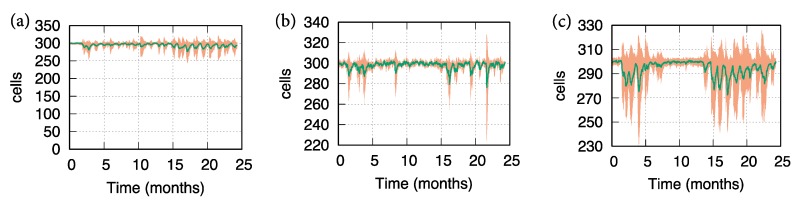
Oligodendrocyte (ODC) dynamics revealed by 20 simulation cases of MS virtual patients. Green lines depict the mean behavior, while the shaded region refers to ± SD. In panel (**a**), simulated MS patients show a slight loss of oligodendrocyte levels. This prediction has been obtained assigning to the factors identified by Rotstein and Montalban input values that are correlated with low severity of MS prognosis. In panel (**b**), simulated MS patients show a medium loss of oligodendrocyte levels. This prediction has been obtained assigning to the factors identified by Rotstein and Montalban input values that are correlated with medium severity of MS prognosis. Finally, in panel (**c**) simulated MS patients show a high loss of oligodendrocyte levels. This prediction has been obtained assigning to the factors identified by Rotstein and Montalban input values that are correlated with high severity of MS prognosis.

**Figure 3 cells-09-00586-f003:**
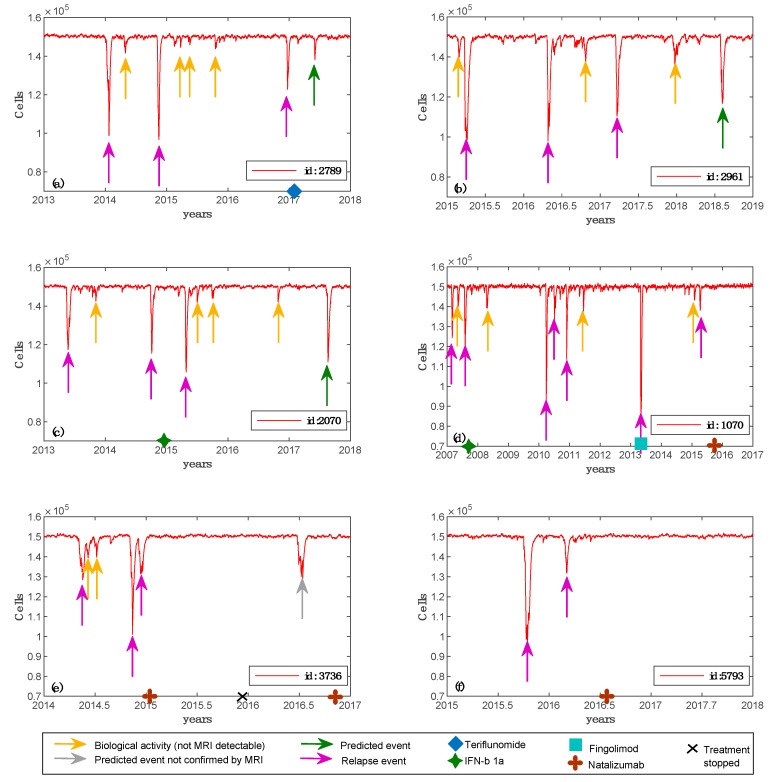
Simulated oligodendrocyte (ODC) dynamics for six patients. For each of the six patients, we depict the best simulation obtained, i.e., the one that parallels real patient data. Specifically, in panel (**a**), virtual patient id 2789 has an age belonging to the range 30–39; MS was initially diagnosed at the age of 33; oligoclonal bands are positive; the patient was two times pregnant; the maximum number of lesions detected by Magnetic Resonance Imaging (MRI) scan was 14; spinal lesions were not detected and teriflunomide was the chosen treatment. In panel (**b**), virtual patient id 2961 has an age belonging to the range 30–39; MS was initially diagnosed at the age of 29; oligoclonal bands are negative; no pregnancy was detected; the maximum number of lesions detected by MRI scan was 65; spinal lesions were not detected and no treatment was chosen. In panel (**c**), virtual patient id 2070 has an age belonging to the range 30–39; MS was initially diagnosed at the age of 31; oligoclonal bands are negative; no pregnancy was detected; the maximum number of lesions detected by MRI scan was 49; spinal lesions were not detected and IFN-β1a was the chosen treatment. In panel (**d**), virtual patient id 1070 has an age belonging to the range 20–29; MS was initially diagnosed at the age of 20; oligoclonal bands are positive; no pregnancy was detected; the maximum number of lesions detected by MRI scan was 82; spinal lesions were detected by MRI scan and IFN-b 1a was the first chosen treatment, then fingolimod and finally natalizumab were administered. In panel (**e**), virtual patient id 3736 has an age belonging to the range 30–39; MS was initially diagnosed at the age of 30; oligoclonal bands are positive; the patient was one time pregnant; the maximum number of lesions detected by MRI scan was 28; spinal lesions were not detected by MRI scan and natalizumab was the chosen treatment. In panel (**f**), virtual patient id 5793 has an age belonging to the range 20–29; MS was initially diagnosed at the age of 26; oligoclonal bands are positive; no pregnancy was detected; the maximum number of lesions detected by MRI scan was 28; spinal lesions were not detected by MRI scan and natalizumab was the chosen treatment.

**Figure 4 cells-09-00586-f004:**
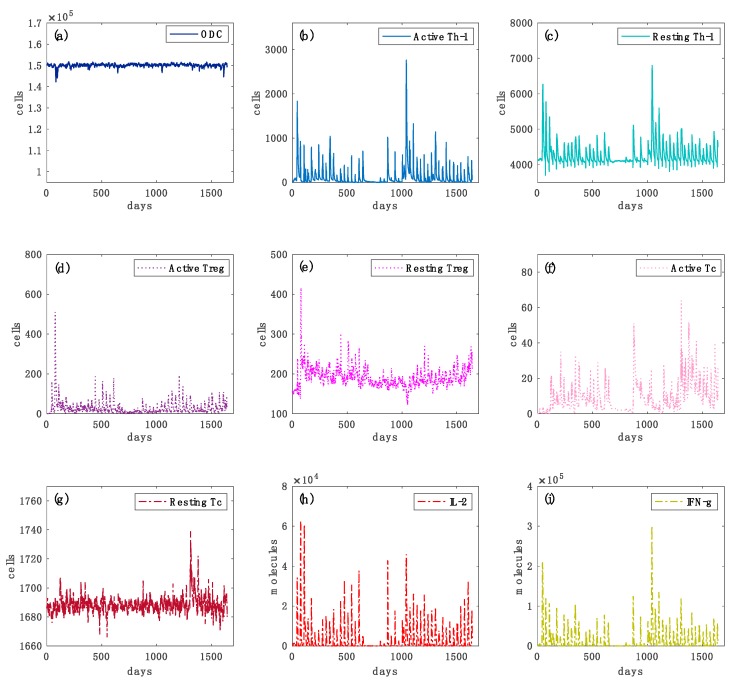
Simulated oligodendrocyte (ODC), Active T helper cells (Active Th-1), Resting T helper cells (Resting Th-1), Active regulatory T cells (Active Treg), Resting regulatory T cells (Resting Treg), Active cytotoxic T cells (Active Tc), Resting cytotoxic T cells (Resting Tc) populations, inflammatory cytokines Interleukin-2 (IL-2) and Interferon-gamma (IFN-g) dynamics for the virtual healthy control patient are respectively depicted in panels (**a**–**i**), respectively. In Figure 4, panel (**a**), the variations in the levels of oligodendrocytes are relatively smaller in healthy controls than those of the diseased cases (Figure 5, panel (**a**)). In panels (**b**,**c**), Th-1 cell populations that could be potentially dangerous in inducing an auto-reactive immune response against oligodendrocyte are kept under control by the regulatory T cells activity. In panels (**d**,**e**), the activation of regulatory T cells and their consequent activity in suppressing any auto-reactive immune response is noticeably increased. In panels (**f**,**g**), active cytotoxic T cells levels are almost inexistent. In panels (**h**,**i**), interleukin 2 (IL-2) and inflammatory interferon gamma (IFN-G) cytokines do not show sufficient levels to induce any inflammatory condition.

**Figure 5 cells-09-00586-f005:**
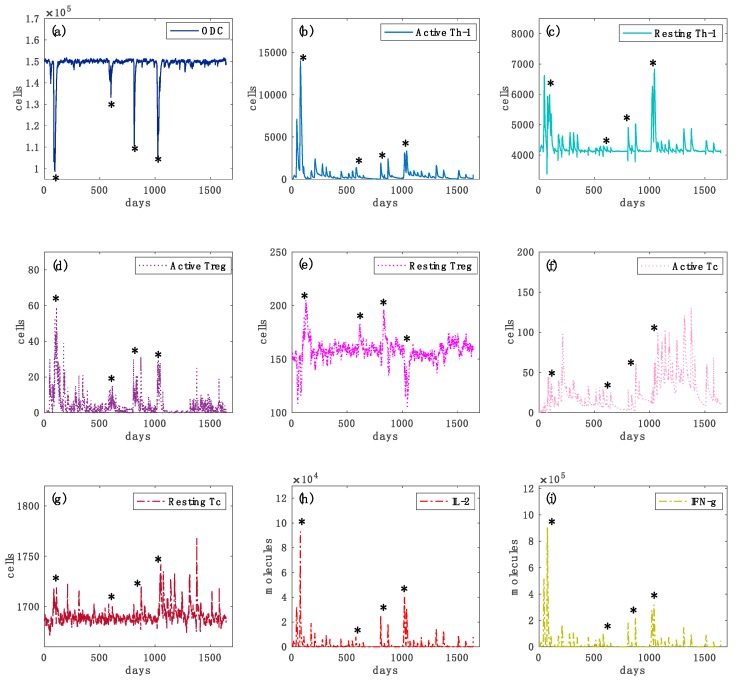
Simulated oligodendrocyte (ODC), Active T helper cells (Active Th-1), Resting T helper cells (Resting Th-1), Active regulatory T cells (Active Treg), Resting regulatory T cells (Resting Treg), Active cytotoxic T cells (Active Tc), Resting cytotoxic T cells (Resting Tc) populations and inflammatory cytokines Interleukin-2 (IL-2) and Interferon-gamma (IFN-g) dynamics for the virtual healthy control patient are respectively depicted in panels (**a**–**i**), respectively. In each panel, star symbols highlight the time-points at which relapses occur, i.e., the time points in which the level of ODC cells goes below the mentioned MRI detectable event threshold. In panel (**a**), one can notice different loss of oligodendrocytes levels that indicate specific relapse events. In panels (**b**,**c**), the levels of active Th-1 cells, which are potentially dangerous in inducing an auto-reactive immune response against oligodendrocytes, are much higher in comparison with the healthy case (Figure 4, panel (**a**)). It is caused by an impaired regulatory T cells activity. In panels (**d**,**e**) the activation of regulatory T cells and their consequent activity in suppressing any auto-reactive immune response is noticeably insufficient. In panels (**f**,**g**), active cytotoxic T cells levels are present, indicating a cytotoxic activity directed to oligodendrocytes. In panels (**h**,**i**), interleukin 2 (IL-2) and inflammatory interferon gamma (IFN-g) cytokines show sufficient levels to induce a strong inflammatory condition.

**Table 1 cells-09-00586-t001:** Cellular and molecular entities implemented in the simulation framework. Initial quantities and half-life values were retrieved from literature in the field [31]. UISS framework takes into account B cells, TH (CD4+ T cells), TC (CD8+ T cells), conventional dendritic cells (CDC), (macrophages (M), plasma B cells (P cells), immunocomplexes (IC), oligodendrocytes (ODC), interferon gamma (IFN-G), interleukins of type x (IL-x), transforming growth factor beta (TGFB), myelin basic proteins (MBP), immunoglobulins class G (IgG) and chemokines (as generic chemokines).

Entity	Initial Quantity per µL (or Cubic Millimeter)	Half-Life (Days or Hours)
B	260	3.3 days
TH	876	3.3 days
TC	434	3.3 days
CDC	351	3.3 days
M	351	3.3 days
P	0	3.3 days
IC	0	4.0 days
ODC	45,000	4.0 days
IFN-G	0	1.6 days
IL-2	0	1.6 days
IL-4	0	1.6 days
IL-10	0	1.6 days
IL-12	0	1.6 days
IL-17	0	1.6 days
IL-23	0	1.6 days
TGFB	0	1.6 days
MBP (myelin basic protein)	0	3 days
IgG	0	23.0 days
Chemokine (generic)	0	3.0 h

**Table 2 cells-09-00586-t002:** Patients data summary. The table shows, for each patient, the clinical information that we translated into simulation terms to reproduce and predict the MS dynamics.

Patient ID	Age Range	MS Age Onset	Oligoclonal Bands (OB)	Pregnancy	Lesional Load (YYYY/MM–Number)	Spinal Lesions (YYYY/MM–Number)	Treatment
1070	20–29	20	yes	no	2007/03–32007/06–5 2010/10–77 2010/12–82 2013/05–59 2015/06–41	2013/05–6 2015/06–7	2007/09: IFN-β1a 2013/05: fingolimod 2015/12: natalizumab
3736	30–39	30	yes	yes, from 2015/11 to 2016/08	2014/09–22 2015/01–28	N/D	2015/02-2015/12: natalizumab 2016/09: natalizumab
2789	30–39	33	yes	yes, from 2012/10 to 2013/07 and from 2015/11 to 2016/08	2013/11–9 2013/12–9 2014/09–9 2017/01–14	N/D	2017/02: teriflunomide
5793	20–29	26	yes	no	2015/08–20 2015/09–22 2016/01–28	N/D	2016/06: natalizumab
2070	30–39	31	no	no	2014/10–23 2015/03–37 2016/01–33	N/D	2014/12: IFN-β1a
2961	30–39	29	no	no	2015/03–49 2016/03–43 2017/04–65	N/D	N/A

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
