# Peer review of "The Potential of Computational Modeling to Predict Disease Course and Treatment Response in Patients with Relapsing Multiple Sclerosis"

_cells, 2020, doi:10.3390/cells9030586_

Round 1

Reviewer 1 Report

The authors addressed all of my concerns. I suggest accepting it.

Author Response

Reviewer actually recommends acceptance. No further revisions are required. Thank you

Reviewer 2 Report

Your responses to the reviewer's comments are noted.

Author Response

Reviewer actually recommends acceptance. No further revisions are required. Thank you

This manuscript is a resubmission of an earlier submission. The following is a list of the peer review reports and author responses from that submission.

Round 1

Reviewer 1 Report

Your paper presents an extremely original computational modeling approach to the evaluation and even prediction of the early course of relapsing MS. While your work is extensive and carefully thought and designed, and indeed the first efffort to produce a computational model to reproduce the course of relapsing/remitting MS, there are certain aspects of the paper that require of clarification or further dicussion. It is perfectly understood yor work is Some aree minor observations.

One of the limitations to assign a number to an ever changing scenario, i.e.number of drugs approved, is that quickly that number is surpassed. If the approved generics for glatiramer acetate by the FDA are counted, the number of approved medications is now 20 (EMA is usually ahead but now is two behind). The approved list include one MAB for PPMS, and 3 medications distinctly approved for active secondary progressive MS. This discussion should be included and updated in this regard. Also, at the international level, there is no such a case as "the most used drugs against MS".  You work around the premise of 'escalation' therapy, that is if a first-line therapy fails, treatment is escalated to a second-line, etc. There is a recent tendency (being studied right now) of 'induction' therapy. Initiating therapy with a high efficacy product swithching later to maintanace with a mild-to-moderate efficay Disease Modifying once the clinical situation estabilizes. This approach is typically being employed at present for highly-active presentation of disease, but it is formally being investigated as a change in therapeutic paradigm regardless of degree of activity of disease. It is not clear how this approach would impact your model.   

Author Response

We thank the reviewer for his positive comment and for highlighting some important clinical aspects. We accordingly revised and updated the number of approved medications for multiple sclerosis at the present time along with the recent tendency of using a sort of “induction therapy”, as kindly suggested by the reviewer.

Reviewer 2 Report

This paper proposes a computational model name UISS (Universal Immune System Simulator) to predict disease course and treatment response in patients with relapsing multiple sclerosis. The authors successfully simulated six patients in different relapsing-remitting courses based on several features like age, sex, presence of oligoclonal bands, therapy and MRI lesion load at onset. This paper is interesting and seems of great significance. However, the following concerns should be addressed to enhance the quality of the paper:

(1) The authors claimed that their model is able to predict the main features and dynamics of the immune system activities especially for patients with relapsing multiple sclerosis. However, is the simulation really true? Are there any ground-truth to compare and confirm to demonstrate the proposed modeling is accurate? Are there any quantitative measurements for the accuracy of the simulation?

(2) The authors should compare their methods with some other state-of-the-art methods. Although they claimed that they are the first to propose such a computational model to reproduce the time course of patients with relapsing multiple sclerosis, some traditional methods (either mathematical model or statistical models) still exist. They should try to compare to convince readers that their model really performs better.

(3) The proposed method seemed to be based on various kinds of features including MRI scanning, therapy, etc. It is questionable whether the cost for the model is worthwhile. What about some patients without all of the required features like MRI features? A good computational model should also consider the cost (e.g., financial cost, time cost, effort cost, etc). The authors should also need to address those cases where not all features for a patient are present.

(4) The link https://combine.dmi.unict.it/ is inaccessible.

(5) The method is not clearly presented. The authors should try to draw a diagram to show the method and specify which steps to finish which sub-goals so that the final simulation or prediction can be achieved.

Author Response

This paper proposes a computational model name UISS (Universal Immune System Simulator) to predict disease course and treatment response in patients with relapsing multiple sclerosis. The authors successfully simulated six patients in different relapsing-remitting courses based on several features like age, sex, presence of oligoclonal bands, therapy and MRI lesion load at onset. This paper is interesting and seems of great significance. However, the following concerns should be addressed to enhance the quality of the paper:

We would like to thank the reviewer for his/her very good comments on the manuscript. Below we provided a point to point answers to the raised questions.

(1) The authors claimed that their model is able to predict the main features and dynamics of the immune system activities especially for patients with relapsing multiple sclerosis. However, is the simulation really true? Are there any ground-truth to compare and confirm to demonstrate the proposed modeling is accurate? Are there any quantitative measurements for the accuracy of the simulation?

The simulations are true. This is highlighted through a comparison between table 2 where a summary of clinical data is shown with Figure 2 in which one can observe that relapse events are conforming to what expected. For example, for patient 2789, three relapse events are reported in table 2; accordingly, the simulation of the same patient in Figure 2 correctly predicted the relapse event. Moreover, the prediction of a new relapse event was confirmed though further MRI scan. We added a sentence strengthening this aspect. Thank you.

(2) The authors should compare their methods with some other state-of-the-art methods. Although they claimed that they are the first to propose such a computational model to reproduce the time course of patients with relapsing multiple sclerosis, some traditional methods (either mathematical model or statistical models) still exist. They should try to compare to convince readers that their model really performs better.

We would like to thank the reviewer for this helpful suggestion. We compared our model with several some other state-of-the-art models, in particular the ones that reproduce the disease progression such as the relapsing-remitting multiple sclerosis (RRMS) behavior. Moreover, we strengthen in what and how our model differs from the ones described. We added accordingly a new paragraph through the text.

(3) The proposed method seemed to be based on various kinds of features including MRI scanning, therapy, etc. It is questionable whether the cost for the model is worthwhile. What about some patients without all of the required features like MRI features? A good computational model should also consider the cost (e.g., financial cost, time cost, effort cost, etc). The authors should also need to address those cases where not all features for a patient are present.

Currently, patients suffering from the first relapse of the disease (even with minor symptoms) are evaluated by a neurologist, who performs a brain MRI that confirms the presence of typical MS lesions. This diagnostic test hence represents the gold standard procedure used to diagnose MS and to track the disease's progression, also due to its non-invasiveness and sensitivity way of imaging the brain. For this reason, we did not take into consideration the possibility that for some patients MRI data was not present. So, the costs of the model do not impact at all during the diagnosis process instead it could drastically reduce any costs deriving through a wrong therapy selection.

(4) The link https://combine.dmi.unict.it/ is inaccessible.

We thank the reviewer for bringing this to our attention. It was due to a misconfiguration in the web server. Now the link works correctly.

(5) The method is not clearly presented. The authors should try to draw a diagram to show the method and specify which steps to finish which sub-goals so that the final simulation or prediction can be achieved.

We thank the reviewer for pointing out this concern. However, now that the simulator is accessible through the working link, one can be aware on the major steps needed to instantiate the simulation in order to achieve the prediction.

Round 2

Reviewer 1 Report

Thank you for providing the changes and concept modification posed by the reviewers.

Reviewer 2 Report

The authors have tried to address most of my concerns raised in the last round of reviews. However, there are some concerns which need further clarifications. In the 1st concern of my last round of review, the authors were suggested to provide quantitative measurements to evaluate how accurate their simulation model is, which, however, seems to have not been addressed. For Comments 2 and 3, I have no idea which paragraphs were added to address how their proposed model outperformed other state-of-the-art methods (it seemed that only some paragraphs were added in the introduction part). And no new contents were added in the results or discussion part. Besides, the discussions on the costs by addressing the 3rd comment should be mentioned in the discussion section to enhance the quality of the paper. For the link, it is still inaccessible.